# Dealing with Headache: Sex Differences in the Burden of Migraine- and Tension-Type Headache

**DOI:** 10.3390/brainsci11101323

**Published:** 2021-10-05

**Authors:** Maria Susanne Neumeier, Heiko Pohl, Peter S. Sandor, Hans Gut, Gabriele S. Merki-Feld, Colette Andrée

**Affiliations:** 1Department of Neurology, University Hospital Zurich, 8091 Zurich, Switzerland; heiko.pohl@usz.ch (H.P.); peter.sandor@zurzachcare.ch (P.S.S.); 2Zurzach Care, 5330 Bad Zurzach, Switzerland; 3Migraine Action Switzerland, 4103 Bottmingen, Switzerland; gut-rella@bluewin.ch (H.G.); migraine_action@vtxmail.ch (C.A.); 4Department of Reproductive Endocrinology, University Hospital Zurich, 8091 Zurich, Switzerland; Gabriele.Merki@usz.ch; 5Department of Pharmaceutical Sciences, University of Basel, 4001 Basel, Switzerland

**Keywords:** sex differences, burden of disease, migraine, tension-type headache, precision medicine

## Abstract

Objective: The aim of this study was to investigate sex differences in the burden of migraine and tension-type headache (TTH). Background: Migraine and TTH are more common in women than in men, with differences in comorbidities, treatment responses, disease-modifying factors, and ictal and interictal burden of disease. Information about sex-related influences on ictal and interictal burden is limited, and an increased understanding is mandatory to provide tailored individual treatment for female and male patients. Methods: Participants answered an online survey based on the EUROLIGHT questionnaire. Inclusion criteria were the consent to participate, complete responses to the diagnostic questions, and information about their sex. Sex differences were investigated using the Mann–Whitney U test or Chi-square test. For detecting factors that influence the burden of disease, we built binary regression models. Results: We included 472 (74.6% female) migraineurs and 161 (59.6% female) participants with TTH. Women with migraine reported significantly more problems in their love lives, more self-concealment, less feelings of being understood by family and friends, more interictal anxiety, a higher pain severity, and more depression and anxiety symptoms than men. For TTH, we did not find significant sex-related differences. A higher headache frequency was the factor that increased the burden of disease in female but not in male migraneurs. Conclusion: The burden of disease was higher in women than men with migraine in many aspects, but not with TTH. Therefore, according to our results, there is a need for sex-specific precision medicine for migraine but not TTH. Controlling the headache frequency with a proper acute or prophylactic treatment and treating comorbid depression and anxiety symptoms is crucial to ease migraine’s burden, especially in women.

## 1. Introduction

Primary headache disorders affect 46% of the general adult population; 11% suffer from migraine and 42% from tension-type headache (TTH) [1]. Both headache types are three times more common in women than in men [2,3].

These sex-related differences do not only regard epidemiology but also comorbidities [4,5], treatment responses [6], and disease-modifying factors [7,8]. Previous studies documented differences between women and men in attack frequency, duration and pain intensity. Furthermore, anxiety and depression symptoms differ for both migraine and TTH [5,9,10,11,12,13]. They suggest itself that these factors, together with the interaction with the healthcare system, influence a person´s disease burden. However, there is also evidence that females’ and males’ reactions to their social environments differ in illnesses [14].

Therefore, we hypothesized that factors influencing the burden of headache might differ between the sexes in many more aspects than those presently known. However, knowledge is scarce as no study compared sex-related influences on the ictal and interictal burden of migraine and TTH.

Recent years have witnessed a growing demand for precision medicine of which, according to Mauvais-Jarvis and co-workers, sex and gender are the foundation [15]. They concluded that sex (as a biological construct) was a genetic modifier of disease pathophysiology, clinical presentation, and treatment response. At the same time, gender (as a social construct) influenced the behavior of and towards patients, thereby modifying and determining their access to medical care. 

This study investigates sex-related differences in the burden of migraine and TTH and analyzes sex-specific aspects, which influences this burden. Finally, we hope that our results will increase the understanding of sex-related differences and contribute to more individual and precise treatments.

## 2. Materials and Methods

### 2.1. Study Design, Inclusion and Exclusion Criteria

The study was internet-based and cross-sectional. Having been informed through posters, journal articles and web pages, participants enrolled from March 2019 to March 2020 and completed the EUROLIGHT questionnaire [16].

We included those who consented to participate and excluded those who provided incomplete responses to the diagnostic questions or did not disclose their biological sex.

As participation was anonymous, the study did not require further ethical approval according to Swiss legislation. 

### 2.2. Outcome Measures

A previously published diagnostic algorithm allowed diagnosing probable migraine, migraine, probable TTH, and TTH [16]. These questions comprised one item that asked participants to rate their pain intensity on an ordinal scale (“not bad”, “bad”, or “very bad”).

Furthermore, the participants answered questions about the ictal and interictal burden with dichotomous answers (yes/no); see Table 1. Three questions assessed interaction with the direct social environment, two questions assessed fear of pain and avoidance behavior, and six questions inquired about the interaction with the healthcare system and diagnostics. 

We estimated anxiety levels and the number of depressive symptoms with the self-rated Hospital Anxiety and Depression scale (HADS), consisting of seven items indicating anxiety (HADS-A) and seven items indicating depressive symptoms (HADS-D). Participants rated their accordance with each item on a 4-point Likert scale (0–3), with a higher score indicating more severe symptoms and a maximum of 21 points for each subscale [17]. The threshold indicating possible anxiety and possible depression was eight; the threshold indicating anxiety or depression was eleven [18]. 

In this work, we differentiated participants based on their biological sex as male and female. We assumed a high overlap between gender and sex among our participants and discussed potential biological and social influences on the burden of disease equally.

### 2.3. Statistical Analysis

We reported categorical variables as frequencies and means with standard deviations and confidence intervals for interval scaled variables. To investigate the influence of the variable “sex” on categorical variables, we used Chi-squared tests; Mann–Whitney U tests analyzed its influence on numerically scaled variables. The burden of disease was investigated for migraine and TTH separately.

Building binary regression models, we analyzed whether age, headache frequency, headache intensity, and the HADS-A and HADS-D influenced the disease burden of migraineurs. We performed these analyses separately in women and men. 

We set the significance level at 0.05 and performed the statistical analysis using IBM SPSS version 27.0.1.0. Missing values were indicated as “not reported” (n.r.).

## 3. Results

A total of 976 participants enrolled in the study, and 633 (46.9%) met the inclusion criteria. We diagnosed migraine and probable migraine in 472 (74.6%) participants, of whom 392 (392/472, 83.1%) were female. One hundred sixty-one participants (161/633, 25.4%), including 96 (96/161, 59.6%) women, met the diagnostic criteria for a tension-type headache (TTH) or probable TTH. In the following, the terms migraine and probable migraine, as well as TTH and probable TTH, were lumped together as migraine and TTH, respectively.

The migraineurs’ mean ages were 40 ± 13 years; female and male participants did not differ significantly in age (39 ± 13 years vs. 41 ± 12 years, *p* = 0.130). Their average headache duration was 9 ± 7 days (women vs. men, 8 ± 8 days vs. 9 ± 7 days, *p* = 0.380) during the last months before completing the questionnaire. The majority of migraineurs were employed, working or studying (424/471, 90%) and married or living with a partner (305/471, 64.8%). 

The mean age of participants with TTH was 39 ± 12 years. Women’s and men’s ages were not statistically significant in their differences (37 ± 13 vs. 41 ± 11, *p* = 0.057). During the last month, the average headache duration was 9 ± 7 days (women vs. men, 8 ± 8 days vs. 5 ± 6 days, *p*= 0.059). The majority of migraineurs were employed, working or studying (155/161; 96.3%) and married or living with a partner (114/159, 89.6%). 

Table 1 specifies the male and female migraineurs’ answers to dichotomous questions. Women reported more problems in their love lives, more self-concealment, and less acceptance by their social environment. They worried more about the next headache episode, had higher pain levels, and had more depression and anxiety symptoms than men. Apart from women receiving a headache diagnosis more often, there was no difference in contact with doctors or diagnostics compared to men. 

Appendix A compares the burden of disease of women and men with TTH; there were no statistically significant sex-related differences.

Table 2 lists the results of the regression models for women suffering from migraine. After correcting for age, headache frequency, attack intensity, anxiety, and depressive symptoms, we found that the headache frequency, intensity, and anxiety symptoms greatly influenced the analyzed aspects of disease burden.

Table 3 summarizes the results of the regression models for men diagnosed with migraine. Compared with women, the investigated covariates had less of an impact on the burden of disease for most questions. A higher headache intensity, and more depression and anxiety symptoms influenced problems in their love lives, and a longer headache duration was associated with worrying more about the next headache episode. 

## 4. Discussion

This study investigated the influence of sex in patients with migraine or TTH on different potential features influencing the burden of the disease. The key findings were (1) marked sex-related differences among migraineurs, with women having a much higher burden of disease, but (2) not among persons with TTH. In addition, (3) the headache frequency emerged as the most crucial influence on the burden of migraine in women, but not in men.

Our study revealed a high prevalence of stigmatization and social isolation, with 39% avoiding telling others about their headache and 17% not feeling understood by family and friends. In a previous study, Lampl et al. [19] found that only 10% of migraineurs did not feel understood by family and friends. This discrepancy suggests cultural differences, as Lampl et al. collected data from several European countries, whereas our sample solely comprises participants from Switzerland.

Zebenholzer et al. [20] investigated factors influencing these aspects of the disease burden. They found that depression increased the risk of avoiding telling others about one’s headaches and increased the risk of feeling lonely and less understood by one’s social environment. These findings fit well with the often-reported loneliness, lack of social support and difficulties in relationships among patients with mental illnesses [21,22]. Therefore, we find it surprising that our results do not confirm the significant influences of depression or anxiety symptoms here. 

At face value, a positive interaction with one’s social environment might be a helpful coping strategy for many patients when experiencing illness or pain. However, headache disorders are mainly characterized by recurring episodes of impairment that might force the patient to interrupt their daily life for several hours or even days and thereby challenge their relationships.

In line with previous studies [19], one third (35.51%) of our participants reported having problems in their love lives due to their headaches, with women reporting more problems than men. Our regression model revealed marked sex-related differences.

A higher headache frequency correlated with problems in participants’ love lives, not feeling understood by their social surroundings, and a higher self-concealment in female patients, but not in men. On the other hand, anxiety and depression symptoms, but not headache frequency, influenced the appearance of problems in the love lives of men. The explanation could be that the loss of time caused by the attacks led to relationship problems in women, whereas the (interictal) affective state had a greater importance in men.

The reasons for women having a higher burden are far from well understood and possibly lie in biology, social sciences, and psychology alike. In contrast to men, women are more likely to seek support from others when ill [14], suggesting the differing interactions with their social environment and the differing significance ratings of others’ support. Considering that the need for support and fear of pain may affect the perceived social support, women might be more likely to deplore its absence. Furthermore, typical gender-related roles, responsibilities, and the distribution of tasks, with women taking care of children and often “managing the daily family life” [23], might be more sensitive to time loss due to a higher attack frequency.

About one third (30.41%) more women than men reported interictal anxiety. Slightly fewer patients (29.76%) avoided “something” in order not to get headaches; here, the sex ratios were balanced. Compared with previous studies, our participants experienced more interictal anxiety, reported more avoidance and no sex-related differences in avoidance [19]. A higher pain intensity was related to a higher probability of interictal anxiety and avoidance in women but not in men. However, in both sexes, a higher headache frequency had a significant influence on interictal anxiety. 

Inconsistent findings for sex-related differences for pain severity were described before [10,24]. Here, in our sample, female migraineurs reported significantly higher pain levels than men.

Previous studies showed a higher comorbidity burden and a bidirectional association with depression, anxiety and migraine, as well as a higher frequency of psychiatric comorbidities in women [9,25]. In line with these findings, female migraineurs in our sample scored higher in both HADS-D and HADS-A. The mean score for both sexes in HADS-D was 7.11 ± 5.28 points, and the mean HADS-A was 8.51 ± 4.61 points. The diagnostic limitations of the HADS did not allow a direct quantitative comparison with former results. However, our results point in the same direction as those from the earlier literature. As psychiatric comorbidities were likely to affect the burden of disease [9], we corrected the regression model for the scores of HADS-A and HADS-D. 

In summary, women experienced a more significant burden of disease, including problems in relationships and their social interactions and love lives, interictal fear, a higher pain intensity, and more depression and anxiety symptoms.

To shed light on the interaction of migraineurs within the healthcare systems, we asked about consultations with primary physicians or headache specialists (neurologist). Compared with earlier studies [26], we found more physician visits, possibly due to different sampling strategies. In women, but not in men, a higher headache frequency correlated with visits of a primary care physician or headache specialist. Interestingly, women received a headache diagnosis from a doctor more often than men did, suggesting that men were underdiagnosed with the “female attributed” disease. In line with our findings, Scher et al. [24] showed that men were less likely than women to receive a migraine diagnosis, possibly due to different illness manifestations and the clinical characteristics of the disease. 

Men and women received medical imaging just as often as each other. However, in women, a younger age and higher pain frequency was associated with a higher probability of receiving an MRI scan of the brain. In addition, women with higher anxiety levels received X-rays of the neck more often. Overall, our results indicated a mostly equal access to healthcare and diagnostics for both sexes. 

Prior studies revealed sex-related differences in tension-type headache, such as higher levels of anxiety and depression in female patients than males [4,27]. In our study, we did not find any significant difference. However, all samples were small and participants were not compared with a control group. Besides, the burden might differ among geographic regions. Thus, more comprehensive studies are required to address sex-related differences in TTH. Nevertheless, our results do not suggest the need to invent or offer sex-specific treatment options to address the disease burden of TTH.

It is observed that the interictal burden in migraine is higher than in TTH [19]. To our knowledge, no previous, comprehensive study investigated the sex-related aspects that influence the burden of disease.

## 5. Limitations

Some limitations need to be mentioned. First, our sample was not representative. Despite a large number of participants, subgroups of the population might not have participated. Second, the sample was not balanced as we included more cases of migraine than TTH and, third, there were more female than male participants. Fourth, we assessed the current headache frequency, while some other questions targeted other time periods. Fifth, HADS was more suitable for screening than diagnosing depression and anxiety, with a tendency to underestimate both [28]. Sixth, we did not correct for multiple testing as this study was exploratory. Therefore, the risk of a Type 1 error might be increased. Seventh, as our study included participants solely living in Switzerland, the results may not be generalized to other countries.

## 6. Conclusions

Our results implied a greater need for sex-specific precision medicine for migraine than for TTH, as we found several sex differences in the burden of disease of migraine but not of TTH. Women experienced a higher burden of disease, especially in relationships and social interactions. For migraine, the predominant factor which increased the burden of disease in women was the headache frequency. Therefore, in everyday practice, sufficient acute medication and prophylaxis were crucial to ease the burden of migraine on women. Anxiety and depression symptoms in migraine were also more pronounced in women, implicating the importance of psychotherapeutic or psychopharmacological treatment. For male migraineurs, we did not find such evident influencing factors on the burden of disease.

As biological, social, or behavioral factors may influence the disease burden, it may be impossible to distinguish the relative importance of sex and gender. Nonetheless, it is essential to consider that the disease burden comprises different aspects that require different approaches in both men and women. 

## Figures and Tables

**Table 1 brainsci-11-01323-t001:** Disease burden of participants with migraine.

	All	Females (%)	Males (%)	Chi-Square	*p*-Value
	472	392 (83.1)	80 (16.9)		
Problems in love life due to headache in the past 3 months					
No	267 (64.49)	211 (61.52)	56 (78.87)	7.739	0.005
Yes	147 (35.51)	132 (38.48)	15 (21.13)		
	58 n.r.	49 n.r.	9 n.r.		
“Do you avoid telling people that you have headaches?”					
No	265 (61.34)	208 (58.26)	57 (76.00)	8.222	0.004
Yes	167 (38.66)	149 (41.74)	18 (24.00)		
	40 n.r.	35 n.r.	5 n.r.		
“Do you feel that your family and friends understand and accept your headaches?”					
No	74 (17.13)	68 (18.99)	6 (8.11)	5.120	0.024
Yes	358 (82.87)	290 (81.01)	68 (91.89)		
	40 n.r.	34 n.r	6 n.r.		
“On that day, was there anything you could not do or did not do because you wanted to avoid getting a headache?”					
No	288 (70.24)	235 (68.51)	3 (79.10)	3.008	0.083
Yes	122 (29.76)	108 (31.49)	14 (20.90)		
	62 n.r.	49 n.r.	13 n.r.		
“On that day, were you anxious or worried about your next headache episode?”					
No	288 (69.59)	232 (67.25)	56 (82.35)	6.142	0.013
Yes	125 (30.41)	113 (32.75)	12 (17.65)		
	59 n.r.	47 n.r.	12 n.r.		
Headache specialist: “Have you had professional advice about your headache in the last year?”					
No	171 (61.29)	152 (61.63)	19 (57.58)	0.218	0.641
Yes	108 (38.71)	94 (38.37)	14 (42.42)		
	193 n.r	146 n.r.	47 n.r		
Primary care doctor: “Have you had professional advice about your headaches in the last year?”					
No	145 (51.97)	131 (53.47)	14 (60.87)	1.367	0.242
Yes	134 (48.03)	115 (46.53)	19 (39.13)		
	193 n.r.	146. n.r.	47 n.r.		
“Has a doctor ever given you a diagnosis for this headache?”					
No	183 (38.94)	140 (35.81)	43 (54.53)	9.588	0.002
Yes	287 (61.06)	251 (64.19)	36 (45.57)		
	2 n.r.	1 n.r	1 n.r.		
“Because of your headaches, have you had an MRI scan in the last year?”					
Yes	76 (39.18)	69 (39.88)	7 (33.33)	0.337	0.561
No	118 (60.82)	104 (60.12)	14 (66.67)		
	278 n.r.	119 n.r.	59 n.r.		
“Because of your headaches, have you had a CT scan in the last year?”					
Yes	32 (16.49)	26 (15.03)	6 (28.57)	2.493	0.114
No	162 (83.51)	147 (84.97)	15 (71.43)		
	278 n.r.	119 n.r.	59 n.r.		
“Because of your headaches, have you had X rays of the neck in the last year?”					
Yes	21 (10.82)	19 (10.98)	2 (9.52)	0.041	0.839
No	173 (89.18)	154 (89.02)	19 (90.48)		
	278 n.r.	119 n.r.	59 n.r.		
“How bad is this headache usually?”					
Not Bad	27 (5.72)	16 (4.08)	11 (13.75)	16.306	<0.001
Bad	261 (55.30)	212 (54.08)	49 (61.25)		
Very Bad	184 (38.98)	164 (41.84)	20 (25.00)		
	0 n.r	0 n.r.	0 n.r.		
	all	females	males		
	mean (SD)	mean (SD)	mean (SD)	Mann–Whitney-U	*p*-value
HADS Depression	7.11 (± 5.28)	7.37 (± 5.33)	5.85 (± 4.83)	11496.500	0.045
	98 n.r.	82 n.r.	16 n.r.		
HADS Anxiety	8.51 (± 4.61)	8.82 (± 4.63)	7.05 (± 4.42)	12291.50	0.004
	99 n.r.	84 n.r.	15 n.r.		

**Table 2 brainsci-11-01323-t002:** Summary of the regression analysis determining the burden of migraine in women.

	Number of Included Cases	Covariates	*p*-Value	Odds Ratio *	95%CI
“During the last 3 months, have your headaches caused difficulties in your love life? “(Yes)	296	Age	0.272	1.011	0.991–1.032
		Headache frequency	<0.001	1.094	1.055–1.135
		Headache intensity	0.141	1.407	0.893–2.218
		HADS-A	0.058	1.087	0.997–1.185
	96 n.r.	HADS-D	0.291	0.961	0.894–1.034
“Do you avoid telling people that you have headaches?” (Yes)	300	Age	0.177	1.013	0.994–1.033
		Headache frequency	<0.001	1.074	1.037–1.112
		Headache intensity	0.785	0.942	0.612–1.450
		HADS-A	0.179	1.058	0.974–1.149
	92 n.r.	HADS-D	0.739	0.988	0.921–1.060
“Do you feel that your family and friends understand and accept your headaches?” (Yes)	300	Age	0.219	0.986	0.964–1.008
		Headache frequency	0.048	0.963	0.927–1.000
		Headache intensity	0.984	0.995	0.592–1.672
		HADS-A	0.943	0.996	0.903–1.100
	92 n.r.	HADS-D	0.332	0.959	0.882–1.043
“On that day, was there anything you could not do or did not do because you wanted to avoid getting a headache?” (No)	295	Age	0.907	0.999	0.978–1.020
		Headache frequency	0.010	0.954	0.920–0.988
		Headache intensity	0.003	0.474	0.292–0.771
		HADS-A	0.027	0.905	0.829–0.989
	97 n.r.	HADS-D	0.279	1.042	0.967–1.122
“On that day, were you anxious or worried about your next headache episode?” (No)	296	Age	0.360	1.010	0.989–1.031
		Headache frequency	<0.001	0.937	0.904–0.971
		Headache intensity	0.004	0.495	0.305–0.803
		HADS-A	0.025	0.905	0.829–0.987
	96 n.r.	HADS-D	0.932	1.003	0.933–1.079
Headache specialist: “Have you had professional advice about your headaches in the last year?” (Yes)	200	Age	0.941	1.001	0.978–1.024
		Headache frequency	0.002	1.065	1.023–1.108
		Headache intensity	0.492	1.209	0.704–2.077
		HADS-A	0.337	0.952	0.862–1.052
	192 n.r.	HADS-D	0.981	1.001	0.921–1.088
Primary care doctor: “Have you had professional advice about your headaches in the last year?” (Yes)	200	Age	0.634	0.995	0.973–1.017
		Headache frequency	0.002	1.064	1.022–1.108
		Headache intensity	0.914	1.029	0.615- 1.722
		HADS-A	0.110	1.082	0.982–1.193
	192 n.r.	HADS-D	0.321	0.960	0.887–1.040
“Has a doctor ever given you a diagnosis for this headache?” (Yes)	300	Age	0.988	1.000	0.981–1.019
		Headache frequency	0.023	1.044	1.006–1.083
		Headache intensity	0.002	2.007	1.287–3.129
		HADS-A	0.544	1.026	0.944–1.116
	92 n.r.	HADS-D	0.231	0.957	0.891–1.028
“Because of your headaches, have you had an MRI scan in the last year?” (Yes)	137	Age	0.030	0.967	0.938–0.997
		Headache frequency	0.046	1.048	1.001–1.098
		Headache intensity	0.174	1.588	0.815–3.093
		HADS-A	0.933	0.995	0.880–1.125
	255 n.r.	HADS-D	0.507	0.966	0.874–1.069
“Because of your headaches, have you had a CT scan in the last year?” (Yes)	137	Age	0.808	0.995	0.958–1.034
		Headache frequency	0.771	1.009	0.951–1.070
		Headache intensity	0.352	0.669	0.287–1.559
		HADS-A	0.178	1.117	0.951–1.312
	255 n.r.	HADS-D	0.127	0.898	0.783–1.031
“Because of your headaches, have you had X rays of the neck in the last year?” (Yes)	137	Age	0.506	1.017	0.968–1.068
		Headache frequency	0.205	1.051	0.973–1.135
		Headache intensity	0.664	1.281	0.419–3.915
		HADS-A	0.041	1.248	1.009–1.542
	255 n.r.	HADS-D	0.767	1.024	0.873–1.202

* An odds ratio above one implies that higher age, headache frequency, headache intensity, scores in HADS-A or HADS-D increase the likelihood of the answer given in brackets. An odds ratio smaller than 1 implies that younger age, lower headache frequency, scores in HADS-A or HADS-D increase the likelihood of the answer given in brackets.

**Table 3 brainsci-11-01323-t003:** Summary of the regression analysis determining the burden of migraine in men.

	Number of Included Cases	Covariates	*p*-Value	Odds Ratio *	95%CI
“During the last 3 months, have your headaches caused difficulties in your love life?” (Yes)	63	Age	0.972	0.999	0.940–1.062
		Headache frequency	0.534	1.031	0.937–1.133
		Headache intensity	0.042	3.521	1.047–11.844
		HADS-A	0.024	0.712	0.530–0.956
	7 n.r.	HADS-D	0.028	1.353	1.034–1.771
“Do you avoid telling people that you have headaches?” (Yes)	63	Age	0.695	0.989	0.934–1.046
		Headache frequency	0.103	1.072	0.986–1.165
		Headache intensity	0.841	1.117	0.378–3.301
		HADS-A	0.401	0.895	0.690–1.160
	7 n.r.	HADS-D	0.094	1.227	0.966–1.558
“Do you feel that your family and friends understand and accept your headaches?” (Yes)	62	Age	0.916	1.005	0.924–1.092
		Headache frequency	0.166	0.925	0.828–1.033
		Headache intensity	0.369	0.487	0.101–2.344
		HADS-A	0.623	1.096	0.762–1.567
	8 n.r.	HADS-D	0.447	0.885	0.645–1.213
“On that day, was there anything you could not do or did not do because you wanted to avoid getting a headache?” (No)	60	Age	0.904	0.996	0.939–1.057
		Headache frequency	0.545	1.030	0.935–1.136
		Headache intensity	0.339	1.751	0.555–5.521
		HADS-A	0.783	1.040	0.787–1.374
	20 n.r.	HADS-D	0.224	0.859	0.671–1.098
“On that day, were you anxious or worried about your next headache episode?” (No)	61	Age	0.688	0.986	0.922–1.055
		Headache frequency	0.007	0.862	0.774–0.961
		Headache intensity	0.442	0.594	0.157–2.242
		HADS-A	0.332	0.872	0.661–1.150
	19 n.r.	HADS-D	0.125	1.253	0.939–1.671
Headache specialist: “Have you had professional advice about your headaches in the last year?” (Yes)	26	Age	0.850	1.008	0.931–1.090
		Headache frequency	0.315	1.072	0.936–1.228
		Headache intensity	0.805	1.194	0.292–4.877
		HADS-A	0.997	0.999	0.715–1.397
	54 n.r.	HADS-D	0.889	1.022	0.751–1.391
Primary care doctor: “Have you had professional advice about your headaches in the last year?” (Yes)	26	Age	0.749	1.013	0.937–1.094
		Headache frequency	0.803	1.018	0.887–1.167
		Headache intensity	0.620	0.710	0.183–2.751
		HADS-A	0.308	0.831	0.583–1.185
	54 n.r.	HADS-D	0.310	1–189	0.851–1.661
“Has a doctor ever given you a diagnosis for this headache?” (Yes)	62	Age	0.111	1.041	0.991–1.095
		Headache frequency	0.243	1.049	0.968–1.136
		Headache intensity	0.133	2.117	0.795–5.632
		HADS-A	0.423	1.089	0.884–1.342
	18 n.r.	HADS-D	0.289	0.902	0.746–1.091
“Because of your headaches, have you had an MRI scan in the last year?” (Yes)	14	Age	0.268	1.081	1.081–0.942
		Headache frequency	0.948	1.007	1.007–0.824
		Headache intensity	0.320	4.181	4.181–0.250
		HADS-A	0.610	1.132	1.132–0.704
	66 n.r.	HADS-D	0.717	0.927	0.927–0.615
“Because of your headaches, have you had a CT scan in the last year?” (Yes)	14	Age	0.762	1.002	0.886- 1.180
		Headache frequency	0.920	0.990	0.807–1.213
		Headache intensity	0.620	0.414	0.013–13.525
		HADS-A	0.327	1.302	0.768–2.207
	66 n.r.	HADS-D	0.544	0.869	0.552–1.368
“Because of your headaches, have you had X rays of the neck in the last year?” (Yes)	14	Age	0.996	50.502	0.000 -
		Headache frequency	0.999	0.534	0.000 -
		Headache intensity	0.997	2.735E+14	0.000 -
		HADS-A	0.997	0.000	0.000 -
	66 n.r.	HADS-D	0.996	12683.366	0.000 –

* An odds ratio above one implies that higher age, headache frequency, headache intensity, scores in HADS-A or HADS-D increase the likelihood of the answer given in brackets. An odds ratio smaller than 1 implies that younger age, lower headache frequency, scores in HADS-A or HADS-D increase the likelihood of the answer given in brackets.

## Data Availability

The data presented in this study are available on request from the corresponding author.

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
