# Peer review of "Dealing with Headache: Sex Differences in the Burden of Migraine- and Tension-Type Headache"

_brainsci, 2021, doi:10.3390/brainsci11101323_

Round 1

Reviewer 1 Report

The authors have performed a very interesting work. 

Methodology is correct

In a web-based design a larger number of subjects could have been desired but the n is enough

I would like have had notice regarding catastrophism scale, not only HADS

Discusion is apropiate and conclusions are correct

I would like to suggest to add some limitations:

  • It is a one-country study and might not be reproducible in other countries
  • Other social and economical variables apart from sex can be related to burden of disease
  • The using of severity-related scales related to migraine (MIDAS or Hit-6) could have expanded the validity of results

Considering that tension-type headache is less frequent that migraine, its under-representation in this study should be argued

Author Response

Point 1: The authors have performed a very interesting work. 

Response 1: Thank you. We appreciate you carefully reviewing the manuscript and the supportive remarks.

Point 2: Methodology is correct.

Response 2: Thank you.

Point 3: In a web-based design a larger number of subjects could have been desired but the n is enough.

Response 3: Thank you for pointing this out. We agree that more participants would have strengthened the confidence in our findings and improved the quality of our results.

Point 4: I would like have had notice regarding catastrophism scale, not only HADS.

Response 4: Thank you for this suggestion. The additional use of the catastrophism scale to the HADS would have added additional insights. We agree that future investigations should include the catastrophism scale.

Point 5: Discussion is apropiate and conclusions are correct.

Response 5: Thank you for proofing the parts of the manuscript. We appreciate your agreement.

Point 6: I would like to suggest to add some limitations: It is a one-country study and might not be reproducible in other countries.

Response 6: We agree with this. Accordingly, we have added the following sentence to the limitations section: “Seventh, as our study includes solely participants living in Switzerland, the results may not be generalized to other countries.”

Point 7: Other social and economical variables apart from sex can be related to burden of disease.

Response 7: You have raised an important point. It would have been interesting to explore this aspect as well. However, our study focused on sex-related differences. We agree that future investigations should include the social and economical variables to further investigate the reasons for these sex differences.

Point 8: The using of severity-related scales related to migraine (MIDAS or Hit-6) could have expanded the validity of results.

Response 8: We agree with this. Further studies should include the severity-related scales related to migraine (MIDAS or Hit-6) for better understanding the disease-related burden.

Point 9: Considering that tension-type headache is less frequent that migraine, its under-representation in this study should be argued

Response 9: Thank you for pointing this out. We agree with this. The underrepresentation of tension-type headache in our sample might be due to the sampling strategy. Therefore, we added the following sentence to the limitations section before: “Second, the sample was not balanced as we included more cases of migraine than TTH and, third, more female than male participants.”

Reviewer 2 Report

First of all, I would like to express my appreciation to the authors for
taking the time and dedication to conduct a review on such a useful topic
as the one chosen. Headache and its burden on the lives of patients are
health problems that are currently treated too often by health services. Although the authors touch on a very important public health problem, I have included some comment about the study that I think would be important to review so that it is clearer and more complete.

  • In the limitations section, the authors indicate that the study has not
    been corrected for multiple testing as this study was exploratory.
    It would be important for the authors to make it clear what implication
    this limitation has when interpreting the results.

Author Response

Point 1: First of all, I would like to express my appreciation to the authors for taking the time and dedication to conduct a review on such a useful topic
as the one chosen. Headache and its burden on the lives of patients are
health problems that are currently treated too often by health services.

Response 1: We thank the reviewer for sharing their expertise and the time spent reviewing the manuscript. We appreciate the supportive comments.

Point 2: Although the authors touch on a very important public health problem, I have included some comment about the study that I think would be important to review so that it is clearer and more complete. In the limitations section, the authors indicate that the study has not been corrected for multiple testing as this study was exploratory. It would be important for the authors to make it clear what implication this limitation has when interpreting the results.

Response 2: Thank you for pointing this out. We agree with this comment. Therefore, we added the following sentence to the limitations section: “Therefore, the risk for Type 1 error might be increased.”

Reviewer 3 Report

The  topic is very interesting. The paper is well written. The language  is understandable . The Methods are described in detail. The results are sound and are discussed in depht. The conclusions are useful from  both a scientific and a clinical point of view.  The paper  will be interesting for readers.

Author Response

Response to Reviewer 3 Comments

The  topic is very interesting. The paper is well written. The language  is understandable . The Methods are described in detail. The results are sound and are discussed in depht. The conclusions are useful from  both a scientific and a clinical point of view.  The paper will be interesting for readers.

Response: Thank you. We appreciate you carefully reviewing the manuscript and the supportive remarks.